# Proliferation dynamics of acute myeloid leukaemia and haematopoietic progenitors competing for bone marrow space

O. Akinduro[1], T. S. Weber[2,3,4], H. Ang[1], M. L. R. Haltalli[1], N. Ruivo[1], D. Duarte[1,5], N. M. Rashidi[1], E. D. Hawkins[1,3,4], K. R. Duffy[2] & C. Lo Celso[1,5]

Leukaemia progressively invades bone marrow (BM), outcompeting healthy haematopoiesis by mechanisms that are not fully understood. Combining cell number measurements with a short-timescale dual pulse labelling method, we simultaneously determine the proliferation dynamics of primitive haematopoietic compartments and acute myeloid leukaemia (AML). We observe an unchanging proportion of AML cells entering S phase per hour throughout disease progression, with substantial BM egress at high levels of infiltration. For healthy haematopoiesis, we find haematopoietic stem cells (HSCs) make a significant contribution to cell production, but we phenotypically identify a quiescent subpopulation with enhanced engraftment ability. During AML progression, we observe that multipotent progenitors maintain a constant proportion entering S phase per hour, despite a dramatic decrease in the overall population size. Primitive populations are lost from BM with kinetics that are consistent with ousting irrespective of cell cycle state, with the exception of the quiescent HSC subpopulation, which is more resistant to elimination.

[1] Department of Life Sciences, Sir Alexander Fleming Building Imperial College London London, SW7 2AZ, UK. [2] Hamilton Institute Maynooth University Maynooth, Co Kildare W23 WK26, Ireland. [3] The Walter and Eliza Hall Institute of Medical Research, Melbourne, VIC 3052, Australia. [4] Department of Medical Biology, The University of Melbourne, Parkville, VIC 3010, Australia. [5] The Francis Crick Institute, 1 Midland Road, London, NW1A 1AT, UK. O. Akinduro and T.S. Weber contributed equally to this work. Correspondence and requests for materials should be addressed to K.R.D. (email: ken.duffy@mu.ie) or to C.L.C. (email: c.lo-celso@imperial.ac.uk)

Haematopoietic cells generate a turnover of billions of blood cells every day. The population-level paradigm for this system is the haematopoietic tree, a hierarchical commitment structure describing progressive amplification and differentiation with rare haematopoietic stem cells (HSCs) at its top[1]. These self-renewing, multipotent cells give rise to a cascade of increasingly lineage-restricted progenitors that are not self-renewing. Steady-state blood cell production heavily relies on proliferation of intermediate progenitor cells, while more primitive populations such as HSCs are known to be relatively quiescent. Consequently, the mechanisms regulating HSC quiescence have been studied in great detail, and dormant HSC subpopulations have been identified[2–5]. In contrast, proliferative HSCs and downstream progenitors have been less well studied, despite their crucial role in maintaining steady-state haematopoiesis and regeneration following injury[6]. Understanding the kinetics of these cells holds vital clues about the regulation of these processes. It has, for example, been determined that HSCs become more proliferative in response to certain stresses, including infection[7–10], but the fate of proliferative stem and progenitor cells under leukaemic stress is unknown.

Leukaemia is a form of cancer that originates from blood lineage cells. As leukaemia invades bone marrow (BM), haematopoiesis decreases to such an extent that patients typically present with symptoms such as anaemia, excessive bleeding or recurrent infections. How leukaemia outcompetes healthy haematopoiesis is only partially understood. It has been established that residual HSCs, both in murine leukaemia models and patients, are still functional in transplantation settings[11–13]. Furthermore, leukaemia-induced changes in BM stroma have been identified at advanced stages of disease[14–17]. Exactly how leukaemia leads to impaired haematopoiesis is yet to be determined, but potential explanations include: (I) a block in differentiation of haematopoietic progenitors[13]; (II) a reduction in stem and progenitor cell production rates[12]; and (III) an increase in apoptosis or emigration. We used the MLL-AF9 mouse model of acute myeloid leukaemia (AML) to better understand the contribution of each of these processes. (I) We determined absolute cell numbers of AML and haematopoietic stem and progenitor cells (HSPCs). (II) We quantified the numbers of AML cells and HSPCs entering S phase per hour. (III) We measured the proportion of apoptotic cells, and the appearance of healthy and malignant cells in blood and spleen.

Measurement of absolute numbers of healthy and apoptotic cells can be obtained by well-established flow cytometry protocols[18, 19]. In contrast, quantification of cell production rates remains challenging. The proliferative behaviour of HSPCs has been studied in vivo through snapshot analysis of cell distribution across cell cycle stages[20, 21], uptake of 5-bromo-9-deoxyuridine (BrdU) or other nucleoside analogues over hours or days[22, 23], and through label retention or dilution[2, 3, 24]. However, cell production rates are not inferable from these methods. Dual pulse-chase nucleoside analogue labelling, pioneered by Wimber and Quastler[25], identifies all cells that have entered into S phase in a given timeframe. We applied it to understand the cellular dynamics underlying AML growth and parallel loss of healthy haematopoiesis. We focused on HSPC populations phenotypically defined by SLAM gating[2, 3, 26–29] and studied the following populations: lineage$^{-/low}$c-Kit$^+$Sca-1$^+$ (LKS) CD150$^+$CD48$^{-/low}$, LKS CD150$^-$CD48$^{-/low}$ and LKS CD48$^+$ cells. For brevity, consistent with multiple recent publications[30–34], we use the terminology HSCs, short term HSCs (ST-HSCs) and multipotent progenitors (MPPs), respectively, for each of these phenotypes. In addition, we measured absolute cell numbers, and proportions of apoptotic cells.

We found that a constant proportion of AML cells entered S phase per hour, and only AML cells in peripheral blood (PB) were less proliferative. We found that in steady-state conditions, MPPs amplify cell numbers more significantly than ST-HSCs or HSCs. The MPP compartment size is, however, comparable with the number of new cells produced every 2 weeks by HSCs and ST-HSCs. We identified that HSCs with the lowest levels of cell surface CD48 expression as the least proliferative in steady-state, and the best engrafters in transplantation. Contrary to expectation, as leukaemia invaded the BM and HSPC compartments became depleted, the fraction of MPPs entering S phase per hour remained largely unchanged. The loss of healthy cells was consistent with neutral ousting from BM by AML irrespective of cell type, with the exception of HSCs with the lowest levels of CD48, which proved more resistant to elimination.

## Results

**AML and HSPC cell counts during AML progression.** GFP$^+$ MLL-AF9-induced blast cells from primary leukaemic mice[35] were injected into non-irradiated secondary recipient mice, which consistently developed high leukaemia burden within 3.5 weeks from injection (Fig. 1a). This approach allowed us to avoid irradiation-related injury of the haematopoietic system. To obtain detailed information on cell number dynamics, at days 12, 15, 18, 21 and 24, BM and spleen were harvested, and AML and non-malignant mononuclear cells (MNC, excluding red blood cells) counts were determined.

AML cell numbers in BM increased exponentially from day 12 to day 18, followed by a period of slower growth in cell numbers over days 21 and 24, which constitute the final stages of the disease (Fig. 1b, left). At day 24, this resulted in $4.87 \times 10^7$ ($\pm 4.6 \times 10^6$ s.e.m) AML cells in BM contained within two tibias, two femurs and two ileac bones of each mouse analysed. BM MNC counts in the same bones started at $1.08 \times 10^8 \pm 3.93 \times 10^6$ cells and remained largely unchanged at homoeostatic levels until day 21. Over days 21–24, as the growth in AML cell numbers was tailing off, MNC counts dropped to approximately a tenth of their starting number (Fig. 1b, right).

During disease progression, the spleen weights were measured, revealing significant enlargement (Fig. 1c) as a result of colonisation by leukaemia cells (Supplementary Fig. 1) combined with extramedullary haematopoiesis. Spleen weight dynamics contrasted with the growth of AML cell numbers in BM, as there was some expansion between days 12 and 18, but a dramatic increase over days 21 and 24, coincident with slower growth of AML cell counts in BM (Supplementary Fig. 1).

Within the MNC compartment, we analysed the cell number dynamics of significant phenotypically defined stem and progenitor cell populations, HSCs (LKS CD150$^+$CD48$^{-/low}$), ST-HSCs (LKS CD150$^-$CD48$^{-/low}$) and MPPs (LKS CD48$^+$) (Supplementary Fig. 2). These cell populations sit at the top of the haematopoietic tree where its topology is typically described as linear, with HSCs giving rise to ST-HSCs, and these in turn to MPPs (Fig. 1d)[26, 29, 30, 32]. HSCs have been further subdivided based on expression of CD34 and functionally by retention of genetic or chemical labels[2, 3, 31, 36]. Lineage tracing experiments have so far considered LKS CD150$^+$ CD48$^{-/low}$ as a single population[6, 30, 32]. LKS CD48$^+$ cells have been further subdivided based on expression of Flk2, CD48 and CD150 to enrich for granulo-monocyte, erythro-megakaryocyte or lymphoid lineage biased MPPs[26, 29, 31].

Downstream of LKS CD48$^+$ cells are populations with a hierarchical relationship that is not yet fully characterised[37–41]. The dynamics of HSC, ST-HSC and MPP cell numbers mimicked those of MNCs as a whole, remaining largely unchanged until growth in AML cell numbers decreased, at which point they dropped dramatically. Interestingly, we detected significant

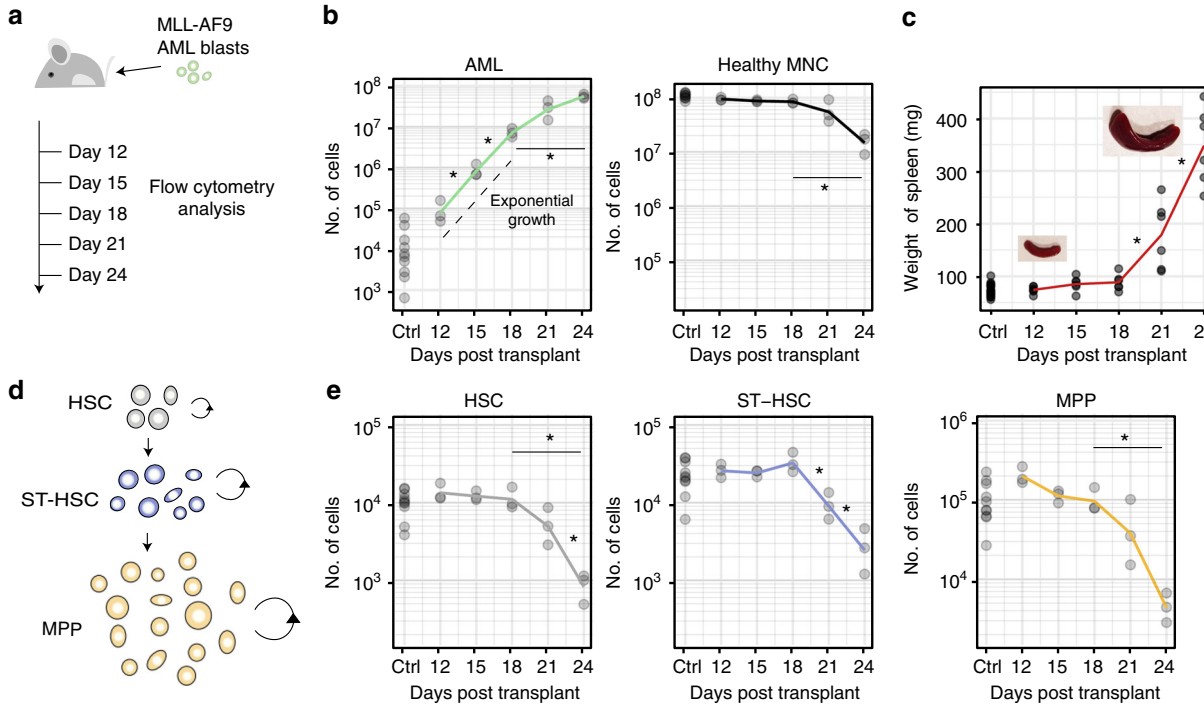

**Fig. 1** Population dynamics of healthy and malignant haematopoietic cells as leukaemia progresses. **a** Cohorts of mice were injected i.v. with 100,000 MLL-AF9 AML blasts at day 0. Control and diseased mice were culled at days 12, 15, 18, 21 and 24, for flow cytometry analysis of BM cells (tibeas, femurs and ileac crest bones). **b** Between days 18 and 24, as AML cell numbers grow in the BM ($p = 0.003$, Welch two-sample $t$-test), mononuclear cell counts decrease significantly ($p = 5.43e{-}04$, Welch two-sample $t$-test). **c** Spleen weights increase following AML cells injection, slowly at first and significantly by day 21 ($p = 0.016$, Welch two-sample $t$-test) and day 24 ($p = 0.001$, Welch two-sample $t$-test). **d** At the apex of the haematopoietic cascade are HSCs, ST-HSCs and MPPs. Cells in each population proliferate (circular arrows) and differentiate into the downstream population (straight arrows). **e** Quantification of HSC (LKS CD150$^+$CD48$^{-/low}$), ST-HSC (LKS CD150$^-$CD48$^{-/low}$) and MPP (LKS CD48$^+$) cell numbers as disease progresses. A significant reduction in population size was observed between days 18 and 24 in all populations ($p = 0.04$ HSCs, $p = 0.03$ ST-HSCs and $p = 0.046$ MPPs, Welch two-sample $t$-test); $n = 15$ control in total and 3 leukaemic mice per time point analysed, apart for **c** where 4–6 spleens/time point are shown. Results from one cohort shown, with equivalent results obtained from two other independent experiments

differences between the cell populations analysed, with ST-HSCs already significantly reduced by day 21, while HSCs and MPPs were more variable at that time (Fig. 1e).

These data were consistent with multiple hypotheses. For example, the AML cell number data were compatible with a decrease in AML proliferation or increase in apoptosis from day 21. They were also compatible, however, with a model where AML proliferation continues unabated, but a lack of space in BM results in cells being displaced into PB. Similarly, HSPC numbers could decrease due to an overall decrease, or even shut down, of healthy cell proliferation and/or increased death. Alternatively, healthy cells could be eliminated by leukaemia through active mechanisms, such as "winning" competition for space. In order to distinguish between these possibilities, we reasoned that we required additional information on proliferation dynamics.

**Identification of cells entering S phase.** Cells undergoing S phase in the presence of BrdU incorporate it into their DNA. Continuous delivery of BrdU results in a consistent pattern of labelling[22] (solid line in Fig. 2a), where the number of BrdU$^+$ cells measured is the sum of all cells that were in S phase at the time of initial delivery plus a decreasing proportion of additional cells over time as BrdU$^-$ cells start DNA synthesis and BrdU$^+$ cells cycle further. Analysis of cells harvested shortly after a pulse of BrdU identifies all cells in S phase in that window, but in order to convert that number into a proportion of cells entering S phase per hour, one would need to divide it by the duration of S phase, which is unknown and variable. The proportion of cells entering

S phase per unit time is the derivative at the origin of the curve depicting the proportion of cells labelled under continuous BrdU administration (Fig. 2a, red dashed line). In the absence of cell cycle arrest and endocycling, this serves as a proxy for the fraction of the population that is proliferating at the time of measurement. This derivative cannot be deduced from a single time-point measurement with a single label-uptake system as one must measure the offset at time zero as well as a second measurement shortly after label administration.

To overcome these limitations, we further developed a short-timeframe, dual pulse label system that was previously established, verified and used in multiple studies, primarily focused on immune cell proliferation[42–47]. It is based on combining BrdU with another thymidine analogue, 5-ethynyl-2′-deoxyuridine (EdU). An initial intravenous EdU pulse was followed 2 h later by intravenous administration of BrdU, and 30 min later cells were harvested and analysed (Fig. 2b). All cells that were in S phase at the time of the initial EdU pulse became EdU$^+$ (blue cells, Fig. 2c, left and middle), while cells that were in S phase during both pulses became EdU$^+$BrdU$^+$ double positive (purple cells, Fig. 2c). Cells that entered S phase in the 2 h between the initial EdU pulse and the BrdU pulse were negative for the EdU label but positive for BrdU label (EdU$^-$BrdU$^+$, orange cells in Fig. 2c). With this method, one obtains a measurement of the proportion of cells in a population entering S phase in a given 2-h time window. By multiplying the proportion of EdU$^-$BrdU$^+$ cells by the overall number of cells, the number of cells in that population that entered S phase in the 2-h window can be calculated.

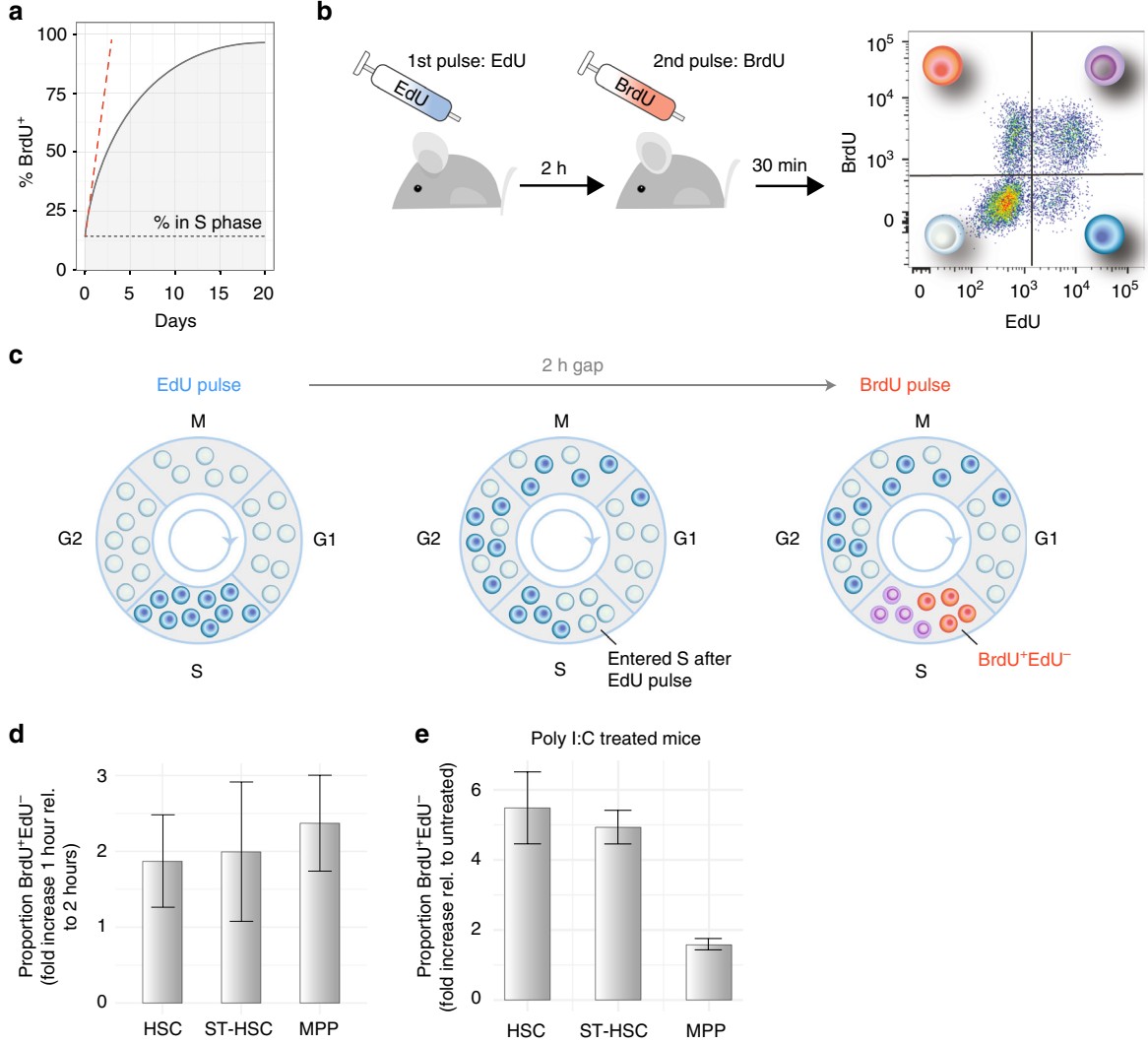

**Fig. 2** EdU and BrdU dual pulse allows measurement of rate of entry into S phase of haematopoietic cell populations. **a** Assuming a constant proportion of cells are in S phase (black dotted line), continuous administration of BrdU alone leads to progressive accumulation of the thymidine analogue in a growing proportion of the cell population (black line). The rate of entry in S phase at the time of the first administration is the derivative at the origin of the curve (dashed red line) and cannot be calculated from a single mouse. **b** Experimental setup: mice were administered EdU first, then BrdU 2 h later, and 30 min later were culled, and haematopoietic cells were analysed by flow cytometry. **c** Diagram representing the dynamics of cell labelling. EdU first labels all the cells in S phase (dark blue). In the 2-h gap, some of these cells progress through G2 and mitosis, while others continue to synthetise DNA (dark blue). In parallel, some unlabelled cells enter S phase (empty circles). BrdU again labels all the cells in S phase, with the following results: cells that entered S phase during the 2-h gap are BrdU single positive (orange circles), cells that were in S phase during both pulses are double positive (purple circles), and cells that terminated S phase prior to the BrdU pulse are EdU single positive (blue circles). **d** Validation of the dual-pulse method. A 2-h gap between pulses results in a 2-fold increase in the number of labelled cells compared to a 1-h pulse (shown is average ± s.e.m; $p = 0.84$ HSC, $p = 0.99$ ST-HSC, $p = 0.61$ MPP, Welch two-sample $t$-test; $n = 3$ mice with 1 h gap, and $n = 3$ mice with a 2 h gap). **e** Dual-pulse method can identify the changes in proliferation rate of cell populations. Mice were administered poly I:C prior to EdU and BrdU, 24 h prior to culling, and the proportions of S phase HSCs, ST-HSCs and MPPs were compared to those of control mice (average ± s.e.m shown; $p = 0.04$ HSC, $p = 0.01$ ST-HSC, $p = 0.07$ MPP, Welch two-sample $t$-test; $n = 3$ control and $n = 3$ poly I:C treated mice)

The interval between the EdU and BrdU pulses was chosen based on two competing factors. For accurate inference, it had to be long enough to capture a sufficiently large number of cells entering S phase. However, it also had to be short enough to ensure that, in the interval between the pulses, cells could neither pass through S phase during the interval without label uptake, as that would result in an EdU⁻BrdU⁻ cell in lieu of an EdU⁻BrdU⁺ cell, nor take up the EdU label, transit through G2/M/G1, and re-enter S phase picking up the BrdU label, as that would result in two EdU⁺BrdU⁺ cells instead of one EdU⁺BrdU⁻ cell. In vitro time lapse microscopy of activated fluorescence ubiquitination cell cycle indicator (FUCCI) lymphocytes indicated a lower bound on G1 of

approximately 2 h and that S/G2/M is typically greater than 4 h[48], guiding our consideration to intervals of 2 h or less.

To confirm the method's accuracy, we analysed HSPC populations of mice that were pulsed with EdU, and received BrdU 1 or 2 h later. As expected, increasing the gap from 1 to 2 h led to a 2-fold increase in the proportions of EdU⁻BrdU⁺ cells for all HSCs, ST-HSCs and MPPs populations (Fig. 2d). We chose a 2-hour labelling period as it increased the number of BrdU single positive labelled cells, and therefore provided increased accuracy for more quiescent populations.

We confirmed that we could detect differences in the rate of S phase entry by examining the mice treated with the immune-

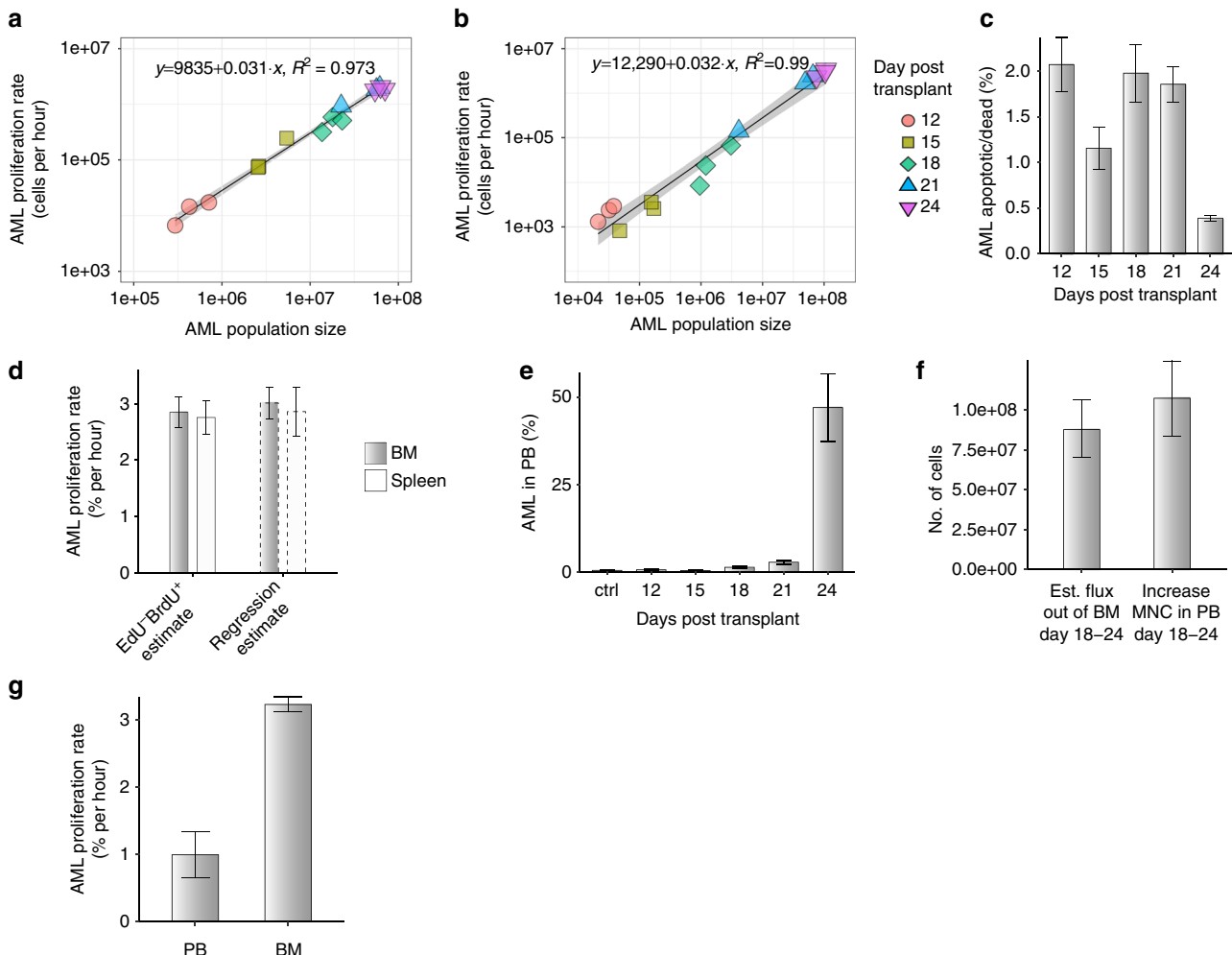

**Fig. 3** Proliferation and BM egression dynamics of AML cells. Number of AML cells entering S phase per hour in BM (**a**) and spleen (**b**) are plotted against absolute numbers of AML cells in tibea, femur and ileac crest bones of mice at multiple time points following injection of AML blasts (colour and shape coded). Black lines are linear regression of the data, with confidence intervals in grey. **c** Very small proportions of apoptotic cells (DAPI and/or Annexin V positive) are detectable within the AML cell population throughout disease progression (average ± s.e.m shown; days 15 and 24, $p = 0.07$, Welch two-sample $t$-test; $n = 3$ mice at each time point). **d** Proportion of AML cells entering S phase per hour estimated for BM (grey bars) and spleen (white bars) using dual-pulse method (left) and exponential regression from data presented in Fig. 1b and Supplementary Fig. 1 (average ± s.e.m; BM vs. spleen EdU⁻ BrdU⁺ estimates $p = 0.8208$, Welch two-sample $t$-test; $n = 9$ mice). **e** The proportion of AML cells in PB (RBC lysed) increases slowly following injection of blasts, and dramatically by the late stages of the disease (average ± s.e.m; $n = 3$ mice per time point). **f** Comparison of estimated number of AML cells exiting BM between days 18 and 24, and measured increase in MNCs in blood during those days (average ± bootstrap std. deviation shown for estimate, average ± s.e.m shown for measured values; estimate, $n = 6$ samples based on measurements presented in Figs. 1b and 3a; measured PB MNC, $n = 6$ mice). **g** Proportion of AML cells entering S phase per hour in blood and BM (average ± s.e.m shown; $p = 0.0003$, Welch two-sample $t$-test; $n = 3$ and 6 mice)

stimulant polyinosinic:polycytidylic acid (poly I:C)[8]. As anticipated, 24 h after poly I:C administration, primitive haematopoietic cells were recruited in the cell cycle, and we detected a 5.4 ± 1.03, 4.9 ± 0.48 and 1.5 ± 0.16-fold increase in the proportion of EdU⁻BrdU⁺ HSCs, ST-HSCs, and MPPs, respectively (Fig. 2 e).

**AML proliferation scales linearly with the population size.** Having established the dual-pulse method, we measured the number of AML cells entering S phase per hour. At every time point, for each mouse we determined both total cell numbers and the number of cells entering S phase in a 2-h window in the BM and spleen. That data established that the number of AML cells entering S phase per hour in both BM (Fig. 3a) and spleen (Fig. 3b) was a fixed fraction of the total number of AML cells. This linear relationship indicated that AML cells maintained

consistent proliferation even when cell numbers in BM were no longer growing exponentially (Fig. 1b, days 21–24).

As AML cells undergo exponential growth in the BM between days 12 and 18 (Fig. 1b), and between days 18 and 24 in the spleen (Supplementary Fig. 1), this afforded us an additional opportunity to confirm the dual label method's accuracy. For a cell population undergoing negligible cell death and growing exponentially, exponential regression enables estimation of the proportion of the population proliferating per unit time. While no method exists to quantitatively determine the number of cells dying per hour in vivo, analysis of apoptotic AML cells by Annexin V and DAPI staining confirmed that these undergo little death (Fig. 3c). Thus, exponential regression from total AML cell numbers provided an estimate that ~3% AML cells enter S phase per hour in both BM and spleen during the exponential growth phase. The dual label measurement provided the same estimate

(Fig. 3d), but required no assumptions on death and could be performed with a single sample harvested at a single time point. Importantly, the method could be applied to non-exponential growth phases, where population size changes could be a consequence of either slower growth or cell loss, or a combination of the two.

**A constant proportion of AML cells enter S phase per hour**. The constant proportion of AML cells entering S phase per hour in BM coupled with the slow down in the growth of AML cell number indicated that from day 18 there was no longer room for all progeny to remain in that organ. We reasoned that this could explain how BM AML cell numbers grew subexponentially (Fig. 1b). Therefore, we hypothesised that a significant number of cells were displaced from the BM once most of that space had been occupied. To test that, we measured AML cell infiltration in PB. Consistent with our hypothesis, PB infiltration increased dramatically during the time when AML growth in the BM was reduced (Fig. 3e). By comparing multiple groups of mice at different stages of BM infiltration, we identified two patterns for the presence of AML cells in the PB where PB infiltration was low when BM infiltration was low (<60%), and then substantially higher when BM infiltration was high (≥60%), (Supplementary Fig. 3).

Based on these data, the number of AML cells produced in BM that are displaced into PB could be estimated (Methods). By day 24, these would constitute a significant proportion of the increase in MNC counts in PB (Fig. 3f). We reasoned that if AML from the BM was a dominant source of AML in the PB, the division rate of PB cells could not account for expansion alone and the proportion of AML cells entering S phase per hour in PB should be lower. Consistent with this hypothesis, the proportion of AML cells entering S phase per hour in PB was approximately one-third of that in BM, at ~1% per hour (Fig. 3g).

**MPPs are major cell amplifiers in steady-state haematopoiesis**. We turned our attention to early HSPCs, first examining steady-state haematopoiesis so that we could better understand the dynamics triggered by leukaemia growth. We acquired sufficient numbers of three phenotypically defined primitive populations (HSCs, ST-HSCs and MPPs) to allow reliable identification of their $BrdU^+EdU^-$, $BrdU^+EdU^+$, $BrdU^-EdU^+$ and $BrdU^-EdU^-$ fractions (Fig. 4a). We also measured the total number of cells in each population in the hind limbs of every mouse analysed (Fig. 4b). As expected, HSCs were least numerous, followed by increasingly more abundant ST-HSCs and MPPs. Knowing the proportion of cells that entered S phase per hour (Fig. 4a) and the number of cells in each population (Fig. 4b), we calculated the number of cells in each population entering S phase per hour (Fig. 4c). Of the three populations, MPPs had by far the greatest number (left panel) and fraction (right panel) of cells entering S phase per hour. The proportion of MPPs entering S phase is substantially higher than that of both HSCs and ST-HSCs, which had similar proportions of cells entering S phase. Thus, the high number of proliferative MPPs is not solely a consequence of the larger size of that population, and the MPP stage is a significant amplifier of cell numbers.

**CD48 level correlates with HSC proliferation and function**. The proliferation and compartment size data allowed us to directly compare HSC and ST-HSC output with MPP cell numbers. Every week LKS CD48$^{-/low}$ cells (HSCs and ST-HSCs combined) produced half as many cells as there were LKS CD48$^+$ cells (MPPs) (Fig. 4d). This finding was consistent with a recent study[6], but did not explain the dormancy or low proliferation rate of HSCs[2–4].

In search of these more quiescent cells, we investigated the distribution of proliferating cells ($EdU^-BrdU^+$, $EdU^+BrdU^+$ and $EdU^+BrdU^-$) within the overall LKS compartment. We noticed that they were particularly rare within the HSC compartment that is most negative for CD48 (Fig. 4e). We therefore re-gated our data based on fluorescence minus one controls to separate, on average, the 25% LKS CD150$^+$CD48$^{-/low}$ cells that had the lowest levels of CD48. Henceforth we denote these as CD48$^{neg}$ HSCs, and the remaining HSCs as CD48$^{low}$ HSCs (Fig. 4f). We found that CD48$^{neg}$ HSCs were turning over at a substantially lower rate than the CD48$^{low}$ HSCs (Fig. 4g). This was consistent with other reports indicating that approximately 20% of LKS CD150$^+$ CD48$^{-/low}$ are label-retaining cells[2] and that not all cells within that phenotypically defined population are functional HSCs in long-term serial transplant settings[26, 29].

To investigate the functional distinction between the CD48$^{neg}$ and CD48$^{low}$ HSCs, we tested their engraftment ability by transplanting 100 mTomato$^+$ cells of either type together with 200 k whole BM haematopoietic cells from wild-type donors into lethally irradiated recipients. The percentage of donor derived cells was measured in the blood of CD48$^{neg}$ and CD48$^{low}$ HSCs recipients at weeks 8, 12 and 20 post transplant, both overall and in the B, T and Myeloid cell compartments (Fig. 4h left panel). BM constitution at week 20 was then analysed according to BM subpopulations (Fig. 4h right panel). Recipients of CD48$^{neg}$ HSCs exhibited consistently higher engraftment at all time points in both PB and BM than recipients of CD48$^{low}$ HSCs, indicating that the quiescent CD48$^{neg}$ population is, indeed, functionally distinct. This result was confirmed by a serial transplantation experiment, which showed that only CD48$^{neg}$ HSCs have appreciable secondary engraftment (Fig. 4i).

**CD48$^{neg}$ HSCs are most resistant to ousting by AML**. In contrast to label flux methods[2–4, 24, 30], the dual label approach is applicable to systems with evolving dynamics such as the haematopoietic shutdown in the BM during leukaemic infiltration. It has been suggested that leukaemia induces a progressive increase of quiescence on residual haematopoietic cells, measured as increasing proportions of cells in G0[12, 13]. One would therefore expect that under leukaemic stress, the proportion of primitive haematopoietic populations entering S phase per hour would progressively decrease. To investigate if this was the case, we analysed the proliferation of residual healthy early HSPCs in the MLL-AF9-driven AML model described earlier (Fig. 1).

We measured the number of CD48$^{neg}$ HSCs, CD48$^{low}$ HSCs, ST-HSCs and MPPs entering S phase per hour (Fig. 5a). For CD48$^{neg}$ HSCs, this was below the limit of detection throughout disease progression. For CD48$^{low}$ HSCs and ST-HSCs, there was no evidence of significant increase or decrease in the proportion of cells entering S phase per hour during disease progression. For MPPs, we found a strong linear correlation with the overall number of MPPs remaining in the BM. This indicated that the proportion of MPPs entering S phase per hour remained unchanged despite their decreasing size.

To better understand how the observed proliferation dynamics fitted with the observed cell loss, we focused on the last 6 days of disease, when all cell populations dramatically decrease in size. Measurement of the average cell numbers for each population (Fig. 5b, bars) established that all cell types were reduced to approximately 10% of their initial size (Fig. 5b, right $y$ axis). In numbers, the MPP compartment decreased most dramatically, but between days 18 and 21, ST-HSCs experienced the greatest proportional decrease and CD48$^-$ HSCs the least. Taken together, these data raised the questions: (1) why did HSCs appear

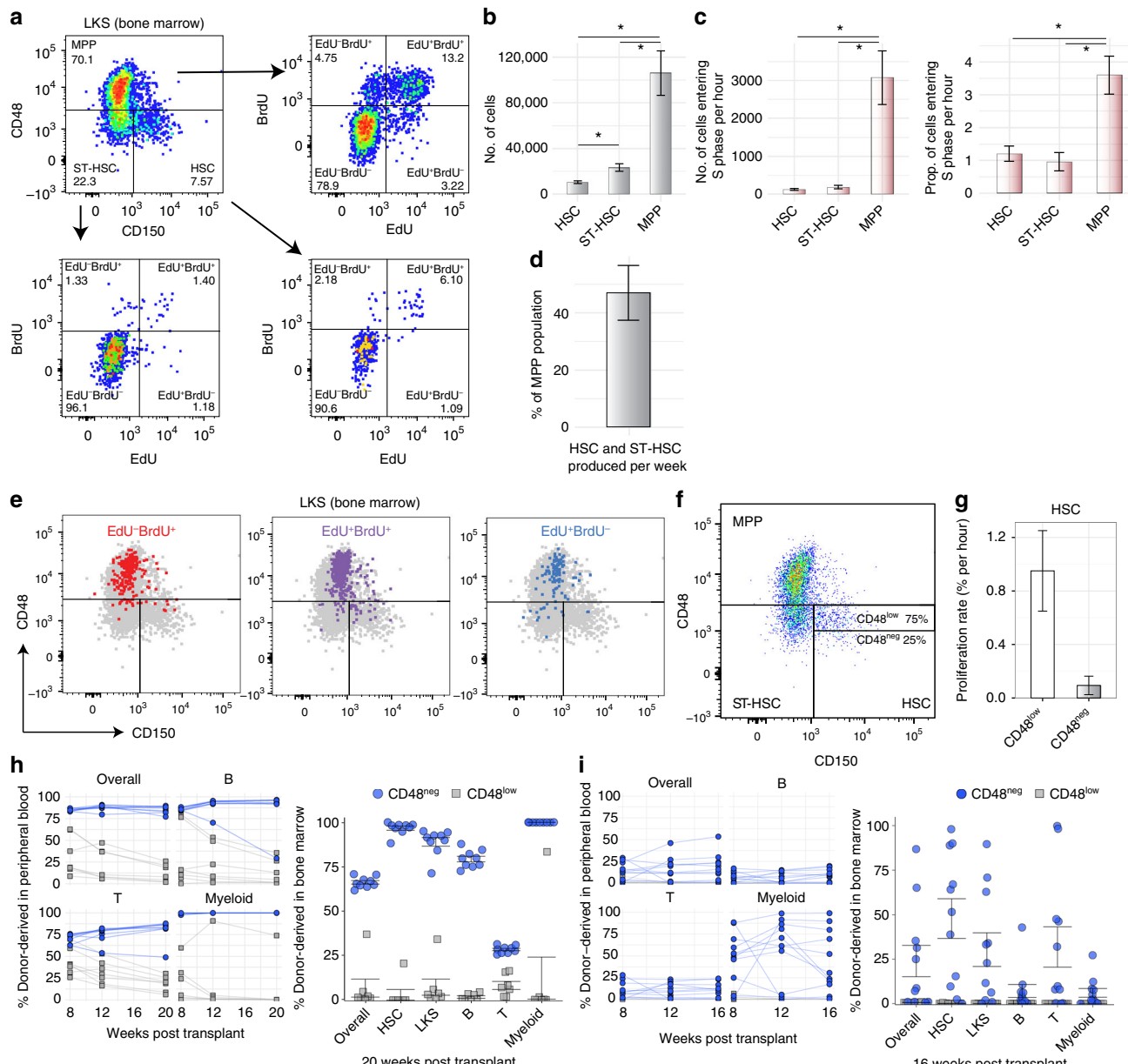

**Fig. 4** EdU-BrdU dual pulse reveals instantaneous rate of entry into S phase of primitive haematopoietic cell populations in steady-state conditions. **a** Representative FACS plot of LKS cells, showing HSC, ST-HSC and MPP populations and their relative FACS plots of EdU and BrdU incorporation patterns. **b** Total number of HSCs, ST-HSCs and MPPs in BM from one hind leg. **c** Number (left) and proportion (right) of HSCs, ST-HSCs and MPPs entering S phase per hour (average ± s.e.m shown; left, $p = 0.252$ HSC to ST-HSC, $p = 0.002$ HSC to MPP, $p = 0.003$ ST-HSC to MPP, Welch's two-sample $t$-test; right, $p = 0.509$ HSC to ST-HSC, $p = 0.004$ HSC to MPP, $p = 0.002$ ST-HSC to MPP, Welch's two-sample $t$-test; $n = 15$ mice analysed). **d** Number of cells produced per week by HSCs and ST-HSCs compared with MPP population size based on the data presented in **b** and **c** (average ± s.e.m. shown). **e** Representative FACS plot of LKS cell population, highlighting the phenotype of EdU−BrdU+, EdU+BrdU+ and EdU+BrdU− cells. **f** Representative FACS plot showing gates separating HSCs with the lowest 25% expression levels of CD48 (CD48neg) and the higher 75% (CD48low). **g** Proportion of CD48low and CD48neg HSCs entering S phase per hour based on EdU and BrdU uptake (average ± s.e.m shown; $p = 0.019$, Welch's two-sample $t$-test; $n = 10$ mice). **h** Transplantation of mTomato+ CD48neg and CD48low HSCs into lethally irradiated recipients. Left panel: time-course of engraftment at weeks 8, 12 and 20 (data, mean ± s.e.m shown; $p < 10^{-4}$ overall, B, T and Myeloid, exact one-way permutation test for mean difference between groups; $n = 9$ and $n = 7$ mice, respectively). Right panel: bone marrow engraftment at week 20. **i** Secondary transplantation of mTomato+ CD48neg and CD48low HSCs into lethally irradiated recipients. Left panel: time-course of engraftment at weeks 8, 12 and 16 (data, mean ± s.e.m shown; $p < 10^{-4}$ overall, B, T and Myeloid, exact one-way permutation test for mean difference between two groups with $n = 11$ mice each). Right panel: bone marrow engraftment at week 16

increasingly quiescent in previous studies[12, 13]; (2) could there be a common mechanism leading to stem and progenitor cell loss from the BM; and (3) are ST-HSCs affected by AML differently compared to the other LKS subpopulations?

To address the first question, we focused on the relative survival of CD48neg and CD48low HSCs, as in steady-state,

CD48neg HSCs were the least proliferative (Fig. 4g). As these cells were being depleted at the slowest rate, we reasoned that this could explain the previously reported increasing proportion of quiescent HSCs[12, 13]. Analysis of CD150 and CD48 expression by flow cytometry in surviving LKS cells at days 18 and 24 confirmed the proportional enrichment of CD48neg HSCs compared with

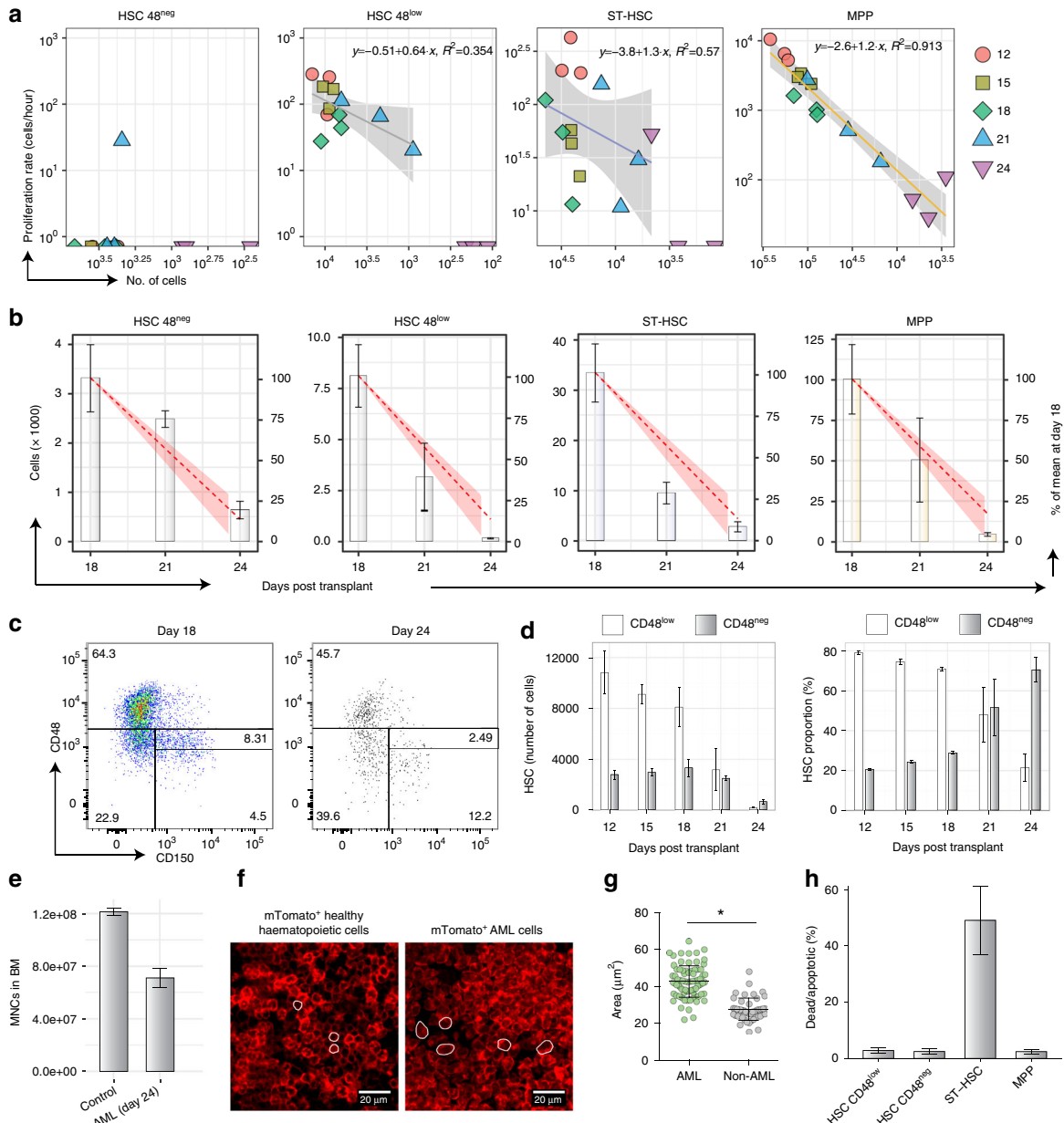

**Fig. 5** Primitive haematopoietic cell populations proliferative dynamics under leukaemic stress. **a** Number of cells entering S phase per hour plotted against number of cells in each of the indicated cell population, colour- and shape-coded per day of analysis following AML blasts injection. Lines show linear regression with 95% confidence intervals shown as grey shadows. **b** Numbers of each analysed population per hind leg, with red dotted line and shaded area indicating the estimated cell loss based on a cell type and cell cycle independent ousting model. **c** Representative FACS plot of LKS cells showing CD48[neg] and CD48[low] gates at days 18 and 24 of disease progression, showing survival of HSCs in the CD48[neg] gate. **d** CD48[neg] and CD48[low] HSC numbers (left) and relative proportions (right) as a function of time (average $\pm$ s.e.m shown). **e** Numbers of mononuclear cells in the hind legs BM of control and advanced leukaemic mice (average $\pm$ s.e.m; $p = 0.012$, Welch's two-sample $t$-test; $n = 2$ control and 3 AML mice from one representative cohort). **f** Histological sections of BM from mice reconstituted with healthy mTmG BM MNC (left) and day 24 mTmG MLL-AF9 blasts (right). Red is membrane-bound tomato signal, while white lines provide illustrative examples of measured cells. **g** Cell area measurements of healthy and AML cells from day 24 of disease ($p < 10^{-4}$, Welch's two-sample $t$-test; $n = 48$ healthy and $n = 78$ AML cells measured from sections from one whole BM chimera and 3 day-24 AML-burdened mice). **h** Proportion of dead or apoptotic (Annexin V and/or DAPI positive) primitive cell populations in mice at an advanced stage of disease, day 24 ($p = 0.68$ for CD48[neg] HSC, $p = 0.24$ for CD48[low] HSC, $p = 0.06$ for ST-HSCs and $p = 0.3$ for MPPs, Welch two-sample $t$-test; $n = 10$ control mice, $n = 3$ leukaemic mice). Throughout the figure, a representative cohort of 15 mice is shown, with 3 analysed at each time point

LKS CD48[low] HSCs at the latter stages of disease (Fig. 5c). Figure 5d shows the evolution of the absolute size of the two HSC subpopulations over time (left panel), where the substantial depletion of CD48[low] HSCs in advance of CD48[neg] HSCs is evident, and results in the relative enrichment of the least proliferative HSCs (right panel).

To explain the declining cell numbers of primitive HSPCs, we hypothesised that all haematopoietic cells may be lost independently of cell type or proliferative state, with the only exception being CD48[neg] HSCs. The sum of all MNCs (healthy and leukaemic populations) in mice at late stages of disease was 40% lower compared to age-matched controls (Fig. 5e). Assuming a

densely packed BM, this suggested that malignant cells are larger, on average, than healthy MNCs. This was confirmed qualitatively by analysing forward scatter of healthy and malignant haemato-poietic cells using flow cytometry (Supplementary Fig. 2), and quantitatively by measuring cell sizes in histological sections of BM from healthy and diseased mice. The latter data also illustrated that the BM was equally packed with haematopoietic cells regardless of steady-state or malignant haematopoiesis (Fig. 5f). The estimated cell size calculated based on MNC numbers and the measured one from histological sections were similar, with AML cells being on average between 1.5 and 2 times bigger than the average healthy cell, based on 1000 bootstrap samples from two control and three AML samples at day 24 (Fig. 5g). We therefore built a simple model in which for every new AML cell that remains in the BM, from day 18 onwards approximately 1.5–2 non-AML cells are randomly ousted without cell type preference. Under these circumstances, the rate of ousting of a cell population is proportional to its size (Methods). The resulting depletion kinetics are presented as red dashed lines in Fig. 5b and, despite the model's simplicity, provide a good explanation of the data. In this model, healthy cells can be lost by both displacement into PB or through death, and the predicted loss of ST-HSCs remains less dramatic than observed.

To understand the interplay of displacement and death as drivers of cell loss, we performed Annexin V and DAPI staining on c-Kit enriched haematopoietic cells harvested from mice at advanced stages of AML progression (Fig. 5h). We observed a modest proportion of apoptotic and dead HSCs and MPPs, equivalent to those observed in healthy control mice (Supplementary Fig. 4), but detected a substantially higher proportion of apoptotic or dead ST-HSCs when compared with other HSPCs (Fig. 5h) and to steady-state BM (Supplementary Fig. 4). The reason why ST-HSCs were observed to be more prone to apoptosis than other HSPC populations, especially at high AML infiltration, remains to be elucidated. A combined effect of death and displacement of ST-HSCs is, however, consistent with their rate of loss being higher than that predicted by elimination alone. The data suggested that elimination from BM rather than cell death may be largely responsible for loss of other HSPCs.

## Discussion

Blood production is maintained by rare HSCs through pro-liferative down-stream multipotent cell types that give rise to lineage committed cell populations. Here, we focused on the proliferation of early haematopoietic cells, which are responsible for blood production maintenance, analysing their rate of entry into S phase in steady-state and under leukaemic stress.

We determined the size of early haematopoietic cell popula-tions in BM as leukaemia progressively infiltrates the BM. We further developed a dual-pulse DNA labelling method that enables in vivo measurement of the proportion of cells entering S phase per hour in multiple cell populations simultaneously. We analysed the malignant component of our system, based on the well-established MLL-AF9-driven murine model of AML. Cell number measurements revealed a decrease in AML growth in BM towards the end of the disease. However, the dual-pulse method established that a constant fraction of AML cells within the BM were entering S phase per hour. Deviation from exponential growth in cell numbers coincided with the appearance of AML cells in the PB, where AML proliferated less vigorously, and suggested a model where AML ultimately runs out of space in the BM. While our data are suggestive of the ousting hypothesis, direct confirmation would require lineage-tracing methodologies. Interestingly, the measured proportion of cells entering S phase

per hour for BM AML cells is within the range observed with leukaemia cells in culture[49]. The reduced rate of entry into S phase of AML cells in PB compared with BM is supportive of current hopes of improving AML outcome through the use of mobilising agents[50–52].

In steady state, we established that every week LKS CD48$^{-/low}$ cells (HSCs and ST-HSCs) produced half as many cells as there were LKS CD48$^{+}$ cells (MPPs), indicating that the most primitive cell population strongly contributes to steady-state haematopoi-esis, in agreement with a recent study[6]. Interestingly, our mea-surements of the proportion of MPP and AML cells entering S phase per hour were not significantly different ($p = 0.38$, Welch's two-sample $t$-test), suggesting that mechanisms other than rapid proliferation are at the root of aggressive expansion observed in AML.

As there is ongoing speculation regarding the level of enrich-ment of long-term HSCs using SLAM gating[2, 3, 27–29] and a feature of our approach is that it allows retrospective re-gating, we investigated the proliferation of the subpopulation of LKS CD150$^{+}$CD48$^{-/low}$ cells identified as having the lowest CD48 expression, which we called CD48$^{neg}$ HSCs. These cells proved to be less proliferative than their LKS CD150$^{+}$CD48$^{low}$ counterpart and thus CD48$^{low}$ HSCs were the more significant contributors to steady-state blood production, consistent with a recently pub-lished study[30]. CD48$^{neg}$ HSCs had greater reconstitution ability than CD48$^{low}$ ones, in agreement with studies showing that quiescence is linked to better engraftment ability[3, 4, 8]. The LKS CD150$^{+}$CD48$^{-/low}$ population has been fractionated based on CD34 expression[36] where CD34$^{-}$ HSCs have been shown to be the most metabolically inactive[3, 4]. Other studies reported that approximately 20% of LKS CD150$^{+}$CD48$^{-/low}$ cells are label retaining[2]. This figure is consistent with our work, which provides a phenotypic identification of the least proliferative HSCs.

Overall, our data confirm that the LKS CD48$^{+}$ population is a critical haematopoietic cell number amplifier, consistent with label retention studies[2, 3]. Our finding that CD48$^{low}$ rather than CD48$^{neg}$ HSCs are more significant contributors to homoeostatic maintenance is in line with the finding that in CD48-null mice HSCs appear more quiescent and haematopoiesis is progressively impaired[53].

As AML cell numbers in BM grow, LKS CD150$^{+}$CD48$^{-/low}$ (HSCs), LKS CD150$^{-}$CD48$^{-/low}$ (ST-HSCs) and LKS CD48$^{+}$ (MPPs) numbers all decreased in concert, suggesting that a process beyond increased HSC quiescence was responsible for their loss. Surprisingly, we found that the fraction of MPPs that entered S phase per hour was near constant throughout disease progression, despite the overall reduction in the population size, while there was no evidence of increased entry into S phase of other HSPCs. This suggests that shut down of healthy haemato-poiesis during leukaemia may be a consequence of a distinct mechanism to the recently proposed one of leukaemia-induced quiescence[12, 13]. Because an increase in cell death as disease progressed was observed only for LKS CD150$^{-}$CD48$^{-/low}$ cells (ST-HSCs), our data led us to propose a model where cells of the haematopoietic system are ousted from the BM by the infiltrating leukaemia irrespectively of proliferative behaviour or phenotypic characteristics, with the exception of LKS CD150$^{+}$CD48$^{-/low}$ cells with the lowest level of CD48 expression, which proved relatively resistant to removal. As these are the least proliferative HSCs, the data reveal a proportional enrichment of apparently quiescent HSCs. This is consistent with earlier studies that identified G0-enriched, engraftment-able HSCs[13] that express high levels of the quiescence inducer Egr3[12] were preserved despite AML cells occupying the majority of BM. Taken together, these and our study raise the questions of how a subpopulation of HSCs sur-vives leukaemia growth longer than any other cell type, and

whether indeed quiescent cells survive or, alternatively, surviving cells are induced into quiescence.

In conclusion, we found the dual-pulse method combined with accurate total cell counts to be essential methods for gaining novel insights in the cellular dynamics underlying the competition between healthy and malignant haematopoiesis. Future studies will indicate whether similar dynamics hold true following chemotherapy treatment, haematopoietic recovery, and disease relapse. Understanding the reason for the differential dynamics of HSPC subpopulations will provide clues for better targeted interventions aimed at restoring healthy haematopoiesis in leukaemia patients.

## Materials and methods

**Mice**. All animal experiments were approved and performed according to the standards of the animal ethics committee at Imperial College London and Sir Francis Crick Institute, and to UK Home Office regulations (ASPA 1986). C57Bl/6 WT mice were purchased from Harlan UK Ltd., and Charles River UK; CD45.1 and mTmG mice were bred and housed at Imperial College London, UK, or provided by the Sir Francis Crick Institute breeding facility (London, UK).

**BrdU and EdU administration**. For cell cycle analysis, EdU and BrdU were used. Both are incorporated during DNA synthesis and, combined, allow analysis of cell cycle kinetics[45]. Pulse-chase timings were optimised to achieve consistent in vivo incorporation of EdU and BrdU, and later detection in the cell populations of interest. EdU (1 mg/mouse) and BrdU (2 mg/mouse) were administered via tail vein injection 2 h apart, unless otherwise stated; 30 min later, mice were culled and the tissue was isolated. Poly-I-poly-C (Sigma) was administered intraperitoneally as a single dose of 100 μg, 24 h prior to culling.

**MLL-AF9 AML model**. Six-week-old CD45.1 or mTmG female mice were BM donors for the generation of AML leukaemia by retroviral transduction of purified granulocyte/monocyte progenitors (GMPs) as previously described[35] using the pMSCV-MLL-AF9-IRES-GFP construct[54]. Transduction of sorted GMPs was performed by plating sorted cells on retrovirus-coated plates in DMEM, 10% FCS, 1% non-essential amino acids, 0.1% β-mercaptoethanol, 0.5% PenStrep, 20 ng/ml SCF, 10 ng/ml IL-3, and 10 ng/ml IL-6 for 72 h. Transduction efficiency was determined by flow cytometry and each primary recipient mouse received 25,000 transduced cells intravenously. Primary recipients were sub-lethally irradiated (two doses of 3 Gy greater than 3 h apart) before transplantation. Primary blasts were harvested from the BM of primary recipients at 8–10 weeks post transplant (infiltration >90%) and banked in liquid Nitrogen. Secondary leukaemias were set up in non-irradiated recipients by injecting 100,000 AML blasts intravenously. Secondary recipients were culled for flow cytometry analysis at the indicated time points during disease development. Total MNC counts in PB were obtained by differential cell count using an Advia 2120 haematology analyser (Siemens Diagnostics).

**Flow cytometry**. All flow cytometry antibodies used are listed in Supplementary Table 1, and all data were acquired on an LSRFortessa flow cytometer (BD BioSciences, CA). For GMP and HSC subpopulations sorting, red blood cells were lysed, then BM cells were stained with a biotin-conjugated lineage cocktail comprising anti-CD3, CD4, CD8, B220, Ter119, Gr1, and CD11b. Lineage positive cells were depleted by using streptavidin-conjugated magnetic beads (Miltenyi Biotech, Germany) and magnetic columns (LD columns/quadromacs holder, Miltenyi Biotech). The lineage negative-enriched cells were stained with streptavidin and antibodies against c-Kit, Sca-1, CD16/32 and CD34 before sorting for GMPs (Lin-c-Kit+Sca-1-CD16/32+CD34hi) using a BD ARIA III.

For the analysis of the stem and progenitor cell compartments, red cell depleted BM or PB cells were stained with the biotin-conjugated lineage antibody cocktail and c-Kit, Sca-1, CD48, CD150 antibodies, followed by streptavidin Pacific Orange (Life Technologies, CA).

BrdU and EdU were detected using a combination of the BrdU Kit (Becton Dickinson) and the Click-iT EdU Kit (Life Technologies): after performing the cell surface antigen stain, the cells were fixed and permeabilised before incubation in the Click-iT EdU reaction. DNase treatment followed and intracellular anti-BrdU stain was performed immediately before analysis by flow cytometry. For analysis of BrdU/EdU incorporation in AML blasts and residual healthy haematopoietic cells, the initial cell surface antigen staining included also anti CD45.1 antibody.

For the detection of apoptotic cells, Annexin V (APC, BB) and DAPI (Life Technologies) staining were performed according to the manufacturers' instructions on red cell lysed (to identify apoptotic AML cells), and red cell lysed, c-Kit enriched BM cells (to identify apoptotic healthy cells) from healthy control and leukaemia-burdened animals. c-Kit enrichment was performed using anti-c-Kit magnetic beads-conjugated antibody and LD columns (Miltenyi) according to the manufacturer's instructions.

Calibrite beads (BD Biosciences) were used to determine the absolute cell numbers in the populations of interest by flow cytometry. These were added to freshly isolated, untreated BM and spleen and red cell lysed blood at a concentration of 2–5% of the total cells in the tube and acquired by flow cytometry, using a white blood cell/MNC gate to exclude erythrocytes from the counts (remaining cells in BM and spleen are referred to as MNCs). Total cell numbers of different cell populations were back calculated based on their frequency in the samples analysed.

**Measuring the proportion of cells entering S phase per hour**. For a given population of interest, let $n(t)$ denote the number of living cells and $n_s(t)$ denote the number of cells in S phase at time $t$. With $e_s[0,t]$ being all cells that enter S phase and $d_s[0,t]$ being all cells that depart S phase in the interval $[0,t]$, then $n_s(t) = n_s(0) + e_s[0,t] − d_s[0,t]$. With $r$ denoting the fraction of cells that enter S phase per unit time, for $t$ sufficiently small $e_s[0,t] = rn(0)t$. With a single label, BrdU, continuously administered from time 0, the number of BrdU+ cells at time $t$ sufficiently small satisfies BrdU+$(t) = n_s(0) + e_s[0,t] = n_s(0) + rn(0)t$. Thus to extract $r$, one would need measurements at two time points as one would determine the derivative of the proportion of BrdU+ cells at $t = 0$, $r = \lim_{t \to 0} \text{BrdU}^+(t)/(n(0)t)$. With two labels, EdU administered at time 0 followed by a gap of length $T$ to second label, BrdU, $r$ can be determined directly. For any $T$ smaller than both the length of S phase and the time to traverse G2/M/G1, one has that EdU−BrdU+$(T) = e_s[0,T] = rn(0)T$ and thus, for $T$ sufficiently small, the fraction of cells entering S phase per unit time can be identified as the fraction of EdU+BrdU+ cells, $r = \text{EdU}^-\text{BrdU}^+ (T)/(n(0)T)$.

**CD48neg and CD48low HSC transplantation**. CD48neg and CD48low HSCs were FACS sorted from the BM of mT/mG donor mice, and 100 were transplanted into lethally irradiated WT recipients (two doses of 5.5 Gy separated by more than 3 h), alongside 200,000 competitor WT BM cells. Baytril antibiotic was administered to recipient mice for 5 weeks post transplant. Multi-lineage engraftment of the recipients was monitored at 8, 12 and 20 weeks post transplant by FACS analysis of PB. Whole BM was harvested from the CD48neg and CD48low HSC primary recipients. Each experimental sample group was used to seed a secondary cohort of lethally irradiated WT recipients, with each mouse receiving 100 pooled tomato+ HSCs (LKS CD150+CD48−/low) and 200,000 competitor WT BM cells. Multi-lineage engraftment was monitored as with the primary recipients.

**AML cell accumulation in PB**. The numbers of MNCs in blood were estimated by measuring MNC counts per ml of blood and based on 2.5 ml total PB per mouse. The difference in numbers between days 18 and 24 was determined by subtracting the average from day 18 from each measurement at day 24.

**Number of AML cells displaced from BM in late stage disease**. The number of AML cells produced but unaccounted for in the BM was computed by assuming linear growth in AML BM cell numbers between days 18 and 21, where there is ~4-fold increase, and between days 21 and 24, where the increase is ~1.9 fold. The average proportion of EdU−BrdU+ AML cells proliferating per hour over days 18–24 was also determined, $p_{AML} = 0.03$. The number of displaced cells was the number of newly created cells in the BM less the number required to explain the number of AML cells in the BM:

$$36 p_{AML}(\text{AML}(18) + 2\,\text{AML}(21) + \text{AML}(24)) − (\text{AML}(24) − \text{AML}(18)).$$

Bootstrap confidence intervals were determined by resampling the data 1000 times, with replacement, and repeating the above computation.

**Estimation of AML and healthy haematopoietic cells size**. Wild-type or Col2.3-GFP mice were lethally irradiated (two times 5.5 Gy, 3 h apart) and reconstituted with whole BM from mT/mG donors to generate chimeras in which the size of healthy mTomato+ BM cells could be measured. Alternatively, non-irradiated wild-type recipients were injected with 100,000 mTomato+ GFP+ MLL-AF9 primary blasts to measure the AML cell size in histological sections. Eight weeks after reconstitution with healthy mTomato+ BM, or at multiple stages of AML disease, tibias and ileac bones were harvested and processed as described before[16]. Briefly, the bones were fixed in periodate-lysine-paraformaldheyde (PLP), immersed in a sucrose gradient and frozen in optimal cutting temperature (OCT) compound (TissueTek). Sections were obtained with a cryostat (Leica) and the Cryojane tape transfer system (Leica Microsystems) and mounted with Prolong Diamond anti-fade (Invitrogen) for analysis. Miroscopic images were acquired with a confocal/two photon hybrid microscope (Zeiss 780 NLO) equipped with a Mai Tai (SpectraPhysics) laser and 488, 561 and 633 nm lasers. The images were analysed using ImageJ/Fiji.

**Statistical analysis**. Data were processed, analysed and visualised using R and MATLAB. Statistical tests were performed as described in figure legends. The outcomes were treated as significant if $p < 0.05$.

**Data availability**. The data that support the findings of this study are available from the corresponding author upon reasonable request.

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

## Acknowledgements

We thank Steve Lane (QIMR Berghofer, Brisbane) for input on establishing the MLL-AF9 AML experimental model; Jane Srivastava, Catherine Simpson and Jess Rowley for support from the ICL Department of Life Sciences Flow cytometry facility; ICL CBS and Crick BRF staff for mouse husbandry; and Peter O'Donovan for critical input on the manuscript. This work was funded by CRUK (grant No. C36195/A1183 to C.L.C.), ERC (grant No. 337066 to C.L.C.), BBSRC (grant No. BB/I004033/1 to C.L.C.), Science Foundation Ireland (grant No. 12IP1263 to K.R.D.), Wellcome Trust (fellowship 105398/Z/14/Z to M.L.H.) and FCT/GABBA (fellowship SFRH/BD/52195/2013 to D.D.).

## Author contributions

T.S.W., E.D.H., K.R.D. and C.L.C. conceived and designed the study. N.M.R. and O.A. established the AML model. N.R., M.L.R.H. and T.S.W. established the dual-pulse labelling methodology for primitive HSPCs and AML. M.L.R.H. performed the engraftment and serial reconstitution experiments. D.D. created and analysed the histological sections. O.A., H.A., M.L.R.H., N.R and C.L.C conducted the core experiments. T.S.W. and K.R.D. performed the statistical analysis. All authors contributed to data interpretation. K.R.D and C.L.C led the writing of the paper, with input from all authors. Note that H.A., M.L.R.H. and N.R. contributed equally to this work.

## Additional information

**Competing interests:** The authors declare no competing financial interests.

