## [Peer Review File · Nature Communications]

Reviewers' comments:

Reviewer #1 (Remarks to the Author):

The manuscript by Akinduro and colleagues nicely describes the kinetics of haematopoiesis during AML disease progression. The authors have used a strategy that allows them to carefully extract dynamics of the expansion of the different AML cell populations during disease progression. Interestingly, this led to the finding that although the overall population of progenitors in the bone marrow decreases at later stages of the disease, their relative proliferation rates do not change. This is important, because it shows that, neutral ousting and not induction of quiescence is the main driver of loss of cells from the bone marrow. The manuscript is well written, experiments are carefully executed and it is well suited for Nature communications. However, I have some concerns that should be addressed.

Major comments

1) The authors have adapted a strategy of short-timescale dual-pulse labelling method to monitor dynamic changes in cell proliferation. Their method is a very elegant and convincing approach to understand the complex kinetics of cell cycle progression in different cell populations. They righteously acknowledge that frequently used single labelling strategies are not informative without the knowledge of unknown parameters like S-phase length. Instead, their short-timescale dual-pulse labelling method can be used to deduct proliferation rates irrespective of these 'unknowns'.

The choice of a two-hour window between the first and second labelling pulse works well for abundant and highly proliferative cell populations. However, this window is suboptimal for populations that are more rare and/or less proliferative, such as the CD48neg HSC population (Fig 4G). This directly results in the inability to analyse proliferation rates of healthy HSCs in diseased bone marrow (Fig 5A). The authors claim that specifically this population is less prone to be eliminated (Fig 5D). However, it is important to determine whether the relative increase in the proportion of CD48neg HSCs is truly caused by decreased loss or other factors contribute to this as well. These cells could, for instance, compensate for the loss of the other healthy cells by increasing their proliferation rates. The authors could either address this experimentally, by increasing the sensitivity of the analysis by enlarging the time between the two labelling pulses. Alternatively, the authors should discuss complementary explanations in the manuscript.

2) Throughout the manuscript the authors 'ignore' the influence of cell death in their analysis. This is based on the observation that: "analysis of apoptotic AML cells by means of Annexin-V and DAPI staining confirmed that these undergo negligible death' (Page 8 and Fig 3C).

Is this effect truly negligible? First, death cells are quickly cleared from the body and therefore a death rate of $\pm 2\%$ is quite significant. Second, it is unclear to me how these analyses were performed. Which cell population from which tissue was measured? Are only intact cells counted or also debris? Which might mean a strong underestimation of the number of apoptotic cells. These analyses should be extended by immunohistochemistry (or if possible intravital microscopy) of the different haematopoietic sites, such as bone & spleen. Such analysis can give a better indication of the real contribution of apoptosis. Or the authors should include the possible effect on the population dynamics in their analysis.

Minor comments:

* Page 4: "As these correspond to cells undergoing cytokinesis, it allows inference of cell production."

This statement is incorrect and should be adapted. The phases of the cell cycle are independently controlled by the different cell cycle checkpoints. Cells that have entered S phase are not irrevocably committed to enter mitosis (controlled by a G2/DNA damage checkpoint). Furthermore, multiple cell types can undergo DNA replication without physical cell division (endocycling). Therefore one cannot state that cells entering S phase corresponds to cells undergoing the final stages of mitosis.

* Page 11: "We found that there was a trend (CD48^{low} HSCs and ST-HSCs) or a strong (MPP) linear correlation with the overall number of cells remaining in the BM (Fig. 5A)"
The described trend for the (ST-)HSCs populations is not convincing (Fig 5A middle graphs). More data points should be added to strengthen this conclusion or the statement should be removed.
* Page 6: There is a typo: "charatcerised"

Reviewer #2 (Remarks to the Author):

Using the well established murine MLL-AF9 AML mouse model with either CD45.1 or mTmG mice, Akinduro and colleagues explore the kinetics of AML cells in comparison to healthy hematopoietic stem and progenitor cells in order to understand how acute leukemia cells take over the bone marrow. Using a dual-pulse labeling method combined with FACS analyses of progenitor compartments and cell number measurements, they show that AML cells divide in a constant fraction with a fixed ratio of cells entering s phase per hour. Kinetics of different stem/progenitor compartments showed MPP to be the most actively cycling compartment but that all normal progenitor compartments decrease irrespective of their cell cycle state except the most primitive long-term HSC.

Although the data presented here is almost solely descriptive and mechanistic investigations are lacking, it is a very elegant study with a simple design and data as they are presented are convincing. The mathematical modelling is a bit complex to understand although it is explained well, but seems sound. The question of how AML takes over the bone marrow and whether normal stem/progenitor cells are actively suppressed or merely displaced is certainly a timely one and an insufficiently answered question. As such, the authors contribute to understanding the kinetics of AML development.

Some queries:

1. AML proliferation drops off at day 24 in this model at which point the spleens become more enlarged and healthy haematopoiesis declines. The authors extrapolate that the bone marrow space is simply used up and AML moves to other organs as well as to PB, where AML cells divide less frequently than in BM. However, there is no real proof that this is the explanation, simply an observation. It seems likely that niche factors are involved, especially as a mechanism leading to less divisions of AML cells in the PB. I realize that experiments in this direction are lengthy and would perhaps constitute a separate work but perhaps the authors could expand on this in the discussion.
2. It is unclear what happens between day 21 and day 24 as to why exponential growth slows. Is this simply the constraints of the bone marrow?
3. Figure 4G and H: As the population of CD48^{neg} HSC is extremely small (there seems to be only one blue dot in the EdU+BrdU⁻ population in 4F) how likely is it that these data are really valid? Was this determined solely by backgating? Would it be possible to add an additional assay to determine the function of these cells? are the authors convinced that this is a real separate population?
4. Figure 5E and F,G and supplementary Figure 2: Cell number is lower in AML marrow at day 24 than in normal haematopoiesis. The authors try to explain this by measuring size of cells and concluding that AML cells are larger than normal cells. Although this is most certainly often the case in primary human AML (as morphology of human BM smears will attest), I find this data the least convincing within this work. Based on the plots presented in suppl. Figure 2, the AML population in the FSC gate does not really seem larger than healthy cells. Is this conclusion based on more data not shown (exemplary plots that better reflect this conclusion)? Also, why should bone marrow in a healthy mouse be as packed as in an AML marrow (Figure 5E)? Given the authors conclusions of ousting of healthy progenitor cells through AML expansion this does not make sense. In addition, in healthy human individuals, marrow cellularity is certainly less than in

full blown AML. Could the authors please clarify and expand on this point. Is Figure 5F and G day 24? Could the use of a chimera explain these observations?

5. Supplementary Figure 3: the infiltration shown corresponds to which days please.

6. I would be interested to know how the author's model would fit with an AML with is refractory or has relapsed and how administration of chemotherapy would affect the model. This is likely to be beyond the scope of the manuscript but would merit consideration/discussion.

Minor points:

1. Numbering of pages (and perhaps line numbering as well) and Figures would be very helpful in reviewing

2. On page 11 of the manuscript, second paragraph Supplementary Figure 3 is stated to show proportion of apoptosis in healthy control mice. This is incorrect and should probably refer to supplementary Figure 4.

3. The introduction is quite long and although it explains the goals very nicely it is in parts redundant to the discussion. On the other hand, the discussion would benefit from some more depth, e.g. how does this data compare or integrate to D. Bonnets data as referenced in #13. This seems pertinent as mechanistic data are lacking.

4. The comment on page 12 that AML cells may have a preference to localise and proliferate in the BM is self evident and not a particular conclusion from the presented data.

We are grateful for your consideration of our submission, for obtaining two informed reviews of it, and for the overall positive and encouraging comments from the reviewers. We were pleased that they recognized the article's contribution, and appreciated their thoughtful comments. A point-by-point response to their feedback can be found below. We believe the alterations improve the paper, and thank the reviewers for their constructive remarks.

Reviewer #1 (Remarks to the Author):

The manuscript by Akinduro and colleagues nicely describes the kinetics of haematopoiesis during AML disease progression. The authors have used a strategy that allows them to carefully extract dynamics of the expansion of the different AML cell populations during disease progression. Interestingly, this led to the finding that although the overall population of progenitors in the bone marrow decreases at later stages of the disease, their relative proliferation rates do not change. This is important, because it shows that, neutral ousting and not induction of quiescence is the main driver of loss of cells from the bone marrow. The manuscript is well written, experiments are carefully executed and it is well suited for Nature communications. However, I have some concerns that should be addressed.

Major comments

1) The authors have adapted a strategy of short-timescale dual-pulse labelling method to monitor dynamic changes in cell proliferation. Their method is a very elegant and convincing approach to understand the complex kinetics of cell cycle progression in different cell populations. They righteously acknowledge that frequently used single labelling strategies are not informative without the knowledge of unknown parameters like S-phase length. Instead, their short-timescale dual-pulse labelling method can be used to deduct proliferation rates irrespective of these 'unknowns'.

The choice of a two-hour window between the first and second labelling pulse works well for abundant and highly proliferative cell populations. However, this window is suboptimal for populations that are more rare and/or less proliferative, such as the CD48neg HSC population (Fig 4G). This directly results in the inability to analyse proliferation rates of healthy HSCs in diseased bone marrow (Fig 5A). The authors claim that specifically this population is less prone to be eliminated (Fig 5D). However, it is important to determine whether the relative increase in the proportion of CD48neg HSCs is truly caused by decreased loss or other factors contribute to this as well. These cells could, for instance, compensate for the loss of the other healthy cells by increasing their proliferation rates. The authors could either address this experimentally, by

increasing the sensitivity of the analysis by enlarging the time between the two labelling pulses. Alternatively, the authors should discuss complementary explanations in the manuscript.

While a longer interval between the pulses would seem desirable, unfortunately, that is not viable within our experimental system. As mentioned in the submitted paper and now more fully explained (revised text below), the gap has to be short enough to ensure that in the interval between the pulses cells can neither pass through S phase without label uptake (which would result in an EdU⁻BrdU⁻ cell in lieu of a EdU⁻BrdU⁺ cell) nor take up the EdU label, transit through G2/M/G1, and re-enter S phase picking up the BrdU label (which would result in two EdU⁺BrdU⁺ cells instead of one EdU⁺BrdU⁻ cell).

While little is known about the length of each phase of the cell cycle of haematopoietic cells in vivo, in vitro time lapse microscopy of strongly activated fluorescence ubiquitination cell cycle indicator (FUCCI) lymphocytes (both B & T cells) (Dowling et al., PNAS, 111(17):6377-82, 2014) revealed a lower bound on G1 of approximately two hours and indicated that S/G2/M is typically greater than four hours, suggesting intervals longer than two hours between EdU and BrdU pulses would run the risk of falling foul of the problems described above.

We confirmed this by including a four-hour interval in preliminary experiments, which we used to determine the optimal gap between the labelling pulses. Consistent with findings by Dowling et al., instead of doubling the number of cells observed to enter S phase, this interval led to fewer cells than expected to be EdU⁻BrdU⁺, and instead a higher proportion of double negative and double positive cells appeared in all populations analysed. While we confirmed that the data was interpretable via an involved mathematical model that required additional caveats and assumptions, we felt that the added layer of obfuscation detracted from the direct and ready interpretability of the data, outweighing any gain.

By way of further explanation, the revised text now reads:

“The two-hour interval between the EdU and BrdU pulses was chosen based on two competing factors. For accurate inference, the interval had to be long enough to capture a sufficiently large number of cells entering S phase, especially in rare, relatively quiescent primitive haematopoietic populations. However, it also had to be short enough to ensure that in the interval between the pulses cells could neither pass through S phase during the interval without label uptake, as that would result in an EdU⁻BrdU⁻ cell in lieu of a EdU⁻BrdU⁺ cell nor take up the EdU label, transit through G2/M/G1, and re-enter S phase picking up the BrdU label, as that would result in two EdU⁺BrdU⁺ cells instead of one EdU⁺BrdU⁻ cell. In vitro time lapse microscopy of activated fluorescence ubiquitination cell cycle indicator (FUCCI) lymphocytes indicated a lower bound on G1 of approximately two hours and that S/G2/M is typically greater than four hours (Dowling et al., PNAS, 2014), guiding our consideration to intervals of two hours or less.”

As increasing the interval was not viable, we have refined the commentary as

detailed below in response to later points by the reviewer (see 'minor comments' responses).

2) Throughout the manuscript the authors 'ignore' the influence of cell death in their analysis. This is based on the observation that: "analysis of apoptotic AML cells by means of Annexin-V and DAPI staining confirmed that these undergo negligible death' (Page 8 and Fig 3C).

Is this effect truly negligible? First, death cells are quickly cleared from the body and therefore a death rate of $\pm 2\%$ is quite significant. Second, it is unclear to me how these analyses were performed. Which cell population from which tissue was measured? Are only intact cells counted or also debris? Which might mean a strong underestimation of the number of apoptotic cells. These analyses should be extended by immunohistochemistry (or if possible intravital microscopy) of the different haematopoietic sites, such as bone & spleen. Such analysis can give a better indication of the real contribution of apoptosis. Or the authors should include the possible effect on the population dynamics in their analysis.

We understand the reviewer's concern on this point, and apologise for an apparent misunderstanding that we think may have led to the comment. In order to quantitatively examine the contribution of cell death to cell number dynamics, knowledge of death rates would be needed. To date, however, this measurement is not achievable with FACS or immunohistochemistry. As a result, we had largely avoided any conclusion that would depend on knowledge of death rates, and, in response to the reviewers comment, we now do so entirely in the revised manuscript.

Using Annexin V and DAPI staining, for AML cells (Fig. 3C, performed on whole bone marrow haematopoietic cell populations harvested from AML-burdened mice) and HSPCs (Supplementary Fig. 4 and Fig. 5H, performed on c-Kit enriched cell population harvested from control and AML-burdened mice, further stained with the indicated antibodies), one obtains identification of cells that are apoptotic or dead, but not a rate of entry into those processes and therefore only relative comparisons can be drawn from this data. Conceptually, it is akin to the issue of only having one label (e.g. BrdU) for proliferation, where here it is the apoptotic process (rather than S phase) whose length is variable and unknown. Unlike proliferation data, however, there is no equivalent of the dual-pulse label approach by which rates can be measured. Regarding the measurements we report, only intact cells were counted. Debris were excluded because they tend to be highly auto-fluorescent, and therefore do not provide a reliable staining. Thus we could focus on early phases of cell death including membrane flipping of phosphatidyl serine (i.e. Annexin V staining).

While our manuscript focused on proliferation, there were three places in the original submission in which we comment on apoptosis.

1) Our first use is in the illustration that the dual pulse method's prediction from single samples is consistent with exponential regression from growth of AML across multiple time-points in multiple animals. The latter method is the usual

technique for inference of a cancer's proliferation rate and its use assumes that death- (and indeed emigration-) rates are zero. We provide evidence in support of that approximation via the same methods that are standard in those studies. Namely, the Annexin V / DAPI staining in Fig. 3C shows lower percentages of apoptotic/dead cells than the healthy populations we investigated (see Supplementary Fig. 4 and Fig. 5H), and so the comparative analysis is taken to imply little death. We are satisfied with that application as it is for confirmation only, and our dual pulse method does not require assumptions on apoptosis rates to provide a measurement of proliferation rate.

2) The second use was in the approximate flux analysis for steady state haematopoiesis leading to the original Fig 4D, where we do a thought experiment considering the levels of differentiation from HSC to ST-HSC to MPPs, assuming death rates to be zero. We appreciate that this is speculative, potentially misleading, and that the picture could be significantly different depending on death rates. Thus, in response to the reviewer's observations, we have removed this entirely from the revised manuscript.

3) The final place is in comparing apoptosis/death of HSPCs in healthy mice and those at the late stages of AML. Here no quantitative inference is made, and only the qualitative observation that apoptosis/death stays largely the same apart from for ST-HSCs, where there is an increase. As this is the standard, qualitative interpretation from data of this sort, we are content that it should be acceptable.

Regarding alternate methods for inference of death rates, we know of none that are feasible and so their development would entail another entire study. Intravital microscopy may allow direct observation of cell death, however the throughput of these analysis is at present too low for rates to be determined.

Minor comments:

* Page 4: "As these correspond to cells undergoing cytokinesis, it allows inference of cell production."

This statement is incorrect and should be adapted. The phases of the cell cycle are independently controlled by the different cell cycle checkpoints. Cells that have entered S phase are not irrevocably committed to enter mitosis (controlled by a G2/DNA damage checkpoint). Furthermore, multiple cell types can undergo DNA replication without physical cell division (endocycling). Therefore one cannot state that cells entering S phase corresponds to cells undergoing the final stages of mitosis.

Thank you for this more precise statement, which we, of course, concur with. We have amended the manuscript, deleting the indicated sentence. We have also altered the description of the dual pulse method to be explicit about its measurement:

"Cells undergoing S phase in the presence of the thymidine analogue BrdU incorporate it into their DNA. Continuous delivery of BrdU results in a consistent pattern of labelling²² (solid line in Fig. 2A), where the number of BrdU⁺ cells

measured is the sum of all cells that were in S phase at the time of initial delivery plus a decreasing proportion of additional cells over time as BrdU⁻ cells start DNA synthesis and BrdU⁺ cells cycle further. Analysis of cells harvested shortly after a pulse of BrdU identifies all cells in S phase in that window, but in order to convert that number into a proportion of cells entering S phase per hour, one would need to divide it by the duration of S phase, which is unknown and likely to be variable. The proportion of cells entering S phase per unit time is the derivative at the origin of the curve depicting the proportion of cells labelled under continuous BrdU administration (Fig. 2A, red dashed line). In the absence of cell cycle arrest and endocycling, this serves as a proxy for the fraction of the population that is proliferating at the time of measurement. This derivative cannot be deduced from a single time-point measurement with a single label-uptake system as one must measure the offset at time zero as well as a second measurement shortly after BrdU administration.”

* Page 11: “We found that there was a trend (CD48^{low} HSCs and ST-HSCs) or a strong (MPP) linear correlation with the overall number of cells remaining in the BM (Fig. 5A)”

The described trend for the (ST-)HSCs populations is not convincing (Fig 5A middle graphs). More data points should be added to strengthen this conclusion or the statement should be removed.

We have deleted those comments from the text, agreeing that they could be misleading, and have changed the abstract to only refer to MPPs. The relevant sections in the revised main text now reads:

“We measured the number of CD48^{neg} HSCs, CD48^{low} HSCs, ST-HSCs and MPPs entering S phase per hour (Fig. 5A). For CD48^{neg} HSCs, this was below the limit of detection throughout disease progression. For CD48^{low} HSCs and ST-HSCs, there was no evidence of significant increase or decrease in proportion of cells entering S phase per hour during disease progression. For MPPs we found a strong linear correlation with the overall number of MPPs remaining in the BM. This indicated that the proportion of MPPs entering S phase per hour remained unchanged despite their decreasing size.”

We have also altered the discussion to more accurately reflect this point:

“Surprisingly, we found that the fraction of MPPs that entered S phase per hour was near constant throughout disease progression, despite the overall reduction in the size of these cell populations, while there was no evidence of increased entry into S phase of other HSPCs.”

* Page 6: There is a typo: “charatcerised”

Thank you, we have corrected it.

Reviewer #2 (Remarks to the Author):

Using the well established murine MLL-AF9 AML mouse model with either CD45.1 or mTmG mice, Akinduro and colleagues explore the kinetics of AML cells in comparison to healthy hematopoietic stem and progenitor cells in order to understand how acute leukemia cells take over the bone marrow. Using a dual-pulse labeling method combined with FACS analyses of progenitor compartments and cell number measurements, they show that AML cells divide in a constant fraction with a fixed ratio of cells entering s phase per hour. Kinetics of different stem/progenitor compartments showed MPP to be the most actively cycling compartment but that all normal progenitor compartments decrease irrespective of their cell cycle state except the most primitive long-term HSC.

Although the data presented here is almost solely descriptive and mechanistic investigations are lacking, it is a very elegant study with a simple design and data as they are presented are convincing. The mathematical modelling is a bit complex to understand although it is explained well, but seems sound. The question of how AML takes over the bone marrow and whether normal stem/progenitor cells are actively suppressed or merely displaced is certainly a timely one and an insufficiently answered question. As such, the authors contribute to understanding the kinetics of AML development.

Some queries:

1. AML proliferation drops off at day 24 in this model at which point the spleens become more enlarged and healthy haematopoiesis declines. The authors extrapolate that the bone marrow space is simply used up and AML moves to other organs as well as to PB, where AML cells divide less frequently than in BM. However, there is no real proof that this is the explanation, simply an observation. It seems likely that niche factors are involved, especially as a mechanism leading to less divisions of AML cells in the PB. I realize that experiments in this direction are lengthy and would perhaps constitute a separate work but perhaps the authors could expand on this in the discussion.

We agree with the reviewer that we cannot provide a definitive proof of splenomegaly being the result of combined influx of AML cells from the bone marrow and local proliferation (reported in Figures 3 D & G), that the recorded reduced proliferation rate of AML cells in PB is likely due to the absence of supportive niches, and that these challenging points will need to be addressed by future studies. As recommended by the reviewer, we have added the following text to the discussion, immediately following our proposal of the BM exit model:

“While our data are suggestive of the veracity of the ousting hypothesis, direct confirmation would require lineage-tracing methodologies.”

2. It is unclear what happens between day 21 and day 24 as to why exponential growth slows. Is this simply the constraints of the bone marrow?

As the proportion of AML cells entering S phase per-hour remains unchanged at ~3% throughout the disease, as does the proportion of apoptotic/dead AML, and day 21 corresponds to both significant BM infiltration and the start of dramatic decline in healthy cell numbers, we believe that is what that the evidence points to. Again, lineage-tracing experiments would be required to definitively prove this point, see point 1.

3. Figure 4G and H: As the population of CD48^{neg} HSC is extremely small (there seems to be only one blue dot in the EdU+BrdU- population in 4F) how likely is it that these data are really valid? Was this determined solely by backgating? Would it be possible to add an additional assay to determine the function of these cells? are the authors convinced that this is a real separate population?

The CD48^{neg} population was not identified by backgating, but by fluorescence minus one controls, which we now clarify in the revision. I.e. we sought out the truly negative CD48 population, and found them to be the bottom ~20-25% of the CD48^{-/low} cells. The amended text is as follows:

“We therefore re-gated our data based on fluorescence minus one controls to separate, on average, the 25% LKS CD150⁺CD48^{-/low} cells that had the lowest levels of CD48.”

The difference in steady state entry to S phase between the CD48^{neg} and the CD48^{low} population is statistically significant, as reported in the legend of the original Fig. 4H (now 4G): p=0.019, Welch’s two sample t-test; n=10 mice.

Regarding function, we have added new data to the revised manuscript. We performed experiments transplanting 100 mTomato⁺ CD48^{neg} and CD48^{low} cells plus 200k WT whole bone marrow haematopoietic cells into 9 and 7 lethally irradiated recipients, respectively. We took a time-course of blood samples at weeks 8, 12 and 20 and investigated BM at week 20, measuring the contribution of mTomato⁺ cells to overall cellularity and to the B, T and Myeloid compartments. The resulting data provide clear evidence that the CD48^{neg} cells engraft best. This indicates that the gating is indeed enriching for cells with distinct properties, both in proliferation rate and engraftment ability.

The new data is reported in the new Fig. 4H and G, and the associated text is:

“To investigate the functional distinction between the CD48^{neg} and CD48^{low} HSCs beyond steady state proliferation, we tested their engraftment ability by transplanting 100 mTomato⁺ cells of either type together with 200k whole bone marrow haematopoietic cells from wild type donors into lethally irradiated recipients. The percentage of donor derived cells was measured in the blood of CD48^{neg} and CD48^{low} HSCs recipients at weeks 8, 12 and 20 post-transplant, both overall and in the B, T and Myeloid cell compartments (Fig. 4H left panel). BM

constitution at week 20 was then analysed according to BM subpopulations (Fig. 4H right panel). Recipients of CD48^{neg} HSCs exhibited consistently higher engraftment at all time-points in both PB and BM than recipients of CD48^{low} HSCs, indicating that the quiescent CD48^{neg} population is, indeed, functionally distinct.”

Ongoing secondary transplants demonstrate that only CD48^{neg} cells are able to serially reconstitute lethally irradiated recipients. We will have week 10-12 data available by the time the current version is reviewed, and we are happy to include them in the final manuscript if the reviewers and editor find it valuable.

4. Figure 5E and F,G and supplementary Figure 2: Cell number is lower in AML marrow at day 24 than in normal haematopoiesis. The authors try to explain this by measuring size of cells and concluding that AML cells are larger than normal cells. Although this is most certainly often the case in primary human AML (as morphology of human BM smears will attest), I find this data the least convincing within this work. Based on the plots presented in suppl. Figure 2, the AML population in the FSC gate does not really seem larger than healthy cells. Is this conclusion based on more data not shown (exemplary plots that better reflect this conclusion)? Also, why should bone marrow in a healthy mouse be as packed as in an AML marrow (Figure 5E)? Given the authors conclusions of ousting of healthy progenitor cells through AML expansion this does not make sense. In addition, in healthy human individuals, marrow cellularity is certainly less than in full blown AML. Could the authors please clarify and expand on this point. Is Figure 5F and G day 24? Could the use of a chimera explain these observations?

The reviewer raises a number of important points that we agree are useful to discuss and address in more detail.

Figure 5F,G indeed show measurements obtained from mice fully infiltrated, i.e. day 24; we now include this detail in the relevant figure legends. We have obtained equivalent results when analysing sections from mice at intermediate levels of infiltration (data not shown). Murine AML cells are truly larger than healthy haematopoietic cells. In addition to Fig. 5F,G, we now show new sample flow cytometry FSC-A plots in Supplementary Figure 2B. Unlike the information in Figure 5F,G, these data are from days 15, 18 and 21 of disease progression, showing that AML cells have consistently larger FSC-A than MNC BM cells. That plot is representative of every AML-burdened mouse we have studied. In order to address any concern the reviewer may have on the specificity of the relative size, for the revised manuscript we performed bootstrap resampling of two control and three AML samples at day 24 getting an interval of AML being 1.5 to 2 times larger than MNC. The revised figure 5B now shows the deductions for this range, and the conclusions remain unchanged.

As additional information for the reviewer, we remark that prior to the analysis shown in Figure 5F,G, we attempted to use flow cytometry to measure the exact size of AML blasts by comparing them to beads of known sizes. The AML cells

always displayed a range of sizes, all bigger than the highest diameter beads in the kit. Because this did not provide a definitive size and did not add to the comparative information we already had, we decided not to show this analysis. Instead we have selected a clearer time-course plot of forward scatter.

Given that human leukaemic marrow is generally hypercellular, we were surprised to see lower total cell numbers in AML-burdened murine bones. We confirmed our finding by assessing cellularity using multiple methods (haemocytometer, not shown; flow cytometry with counting beads, shown). The most plausible explanation that we find is that healthy human BM is less packed than murine bone marrow, especially because it contains many adipocytes. Therefore, it is likely that in humans AML cells can occupy some of the non-utilised space in bone marrow. Instead, murine healthy bone marrow tissue is more heavily packed, and for example contains little adipocytes (the only exception being the tail bones, which we did not study here). Therefore, it is conceivable that AML cells must eliminate healthy cells to make space for themselves.

No bone marrow cellularity data reported in the paper were obtained from chimeras. However, we found that the size of haematopoietic cells in donor and chimeric mice to be the same (data not shown).

5. Supplementary Figure 3: the infiltration shown corresponds to which days please.

Thank you for the suggestion. Measurements are shown for days 11, 12, 14, 15, 17, 18, 21, 22, 23, 24 and 26, and that information is now displayed in the revised Supplementary Figure 3 as a color-coding in a scatter plot.

6. I would be interested to know how the author's model would fit with an AML with is refractory or has relapsed and how administration of chemotherapy would affect the model. This is likely to be beyond the scope of the manuscript but would merit consideration/discussion.

While they are beyond the scope of the present study, those are fine questions, and one presumes that the dual pulse method would be revealing for them. As suggested, we have added a comment to that effect in the discussion:

“Future studies will indicate whether similar dynamics hold true following chemotherapy treatment, haematopoietic recovery, and disease relapse.”

Minor

points:

1. Numbering of pages (and perhaps line numbering as well) and Figures would be very helpful in reviewing

We agree with the reviewer and have added page and line numbers in the revision, as well as Figure numbers.

2. On page 11 of the manuscript, second paragraph Supplementary Figure 3 is stated to show proportion of apoptosis in healthy control mice. This is incorrect and should probably refer to supplementary Figure 4.

We are grateful for pointing that out that typo, which has been corrected in the revision.

3. The introduction is quite long and although it explains the goals very nicely it is in parts redundant to the discussion. On the other hand, the discussion would benefit from some more depth, e.g. how does this data compare or integrate to D. Bonnets data as referenced in #13. This seems pertinent as mechanistic data are lacking.

We have cut some material from the introduction and the discussion to avoid repetition, and have added a more extensive discussion regarding the Bonnet and Cheng data, as well as the other matters suggested by the reviewer. Specific text added in the discussion:

“This is consistent with earlier studies that identified G0-enriched, engraftment-able HSCs¹³ that express high levels of the quiescence inducer Egr3¹² were preserved despite AML cells occupying the majority of bone marrow. Taken together, these and our study raise the questions of how a subpopulation of HSCs survives leukaemia growth longer than any other cell type, and whether indeed quiescent cells survive or, alternatively, surviving cells are induced into quiescence.”

4. The comment on page 12 that AML cells may have a preference to localise and proliferate in the BM is self evident and not a particular conclusion from the presented data.

We concur and have deleted “and suggests that AML cells may have a preference to localise and proliferate within the BM.”

REVIEWERS' COMMENTS:

Reviewer #1 (Remarks to the Author):

The authors have carefully answered my questions. Furthermore, when appropriate, they have adapted the manuscript and my concerns have all been addressed. Therefore I now fully support publication of this manuscript in Nature Communications.

Reviewer #2 (Remarks to the Author):

In their revised version of the manuscript, the authors have adequately addressed all of the reviewer's concerns and have added new experimental data, including new transplantation experiments with CD48^{low} vs CD48^{neg} cells which add more evidence to their claims. The authors offer to include pending data on secondary transplants of these CD48^{neg} cells which I believe would further strengthen the manuscript and add to its significance.

We are pleased that the reviewers were satisfied with the revised paper. The alterations made to address their concerns improved the paper, and we thank them for their constructive remarks. Please find below our response to their last remaining remarks.

Reviewer #1 (Remarks to the Author):

The authors have carefully answered my questions. Furthermore, when appropriate, they have adapted the manuscript and my concerns have all been addressed. Therefore I now fully support publication of this manuscript in Nature Communications.

Thank you.

Reviewer #2 (Remarks to the Author):

In their revised version of the manuscript, the authors have adequately addressed all of the reviewer's concerns and have added new experimental data, including new transplantation experiments with CD48^{low} vs CD48^{neg} cells which add more evidence to their claims.

The authors offer to include pending data on secondary transplants of these CD48^{neg} cells which I believe would further strengthen the manuscript and add to its significance.

In response to this latter request, which we were happy to address, we have added secondary transplant data from weeks 8, 12 and 16. These appear in Figure 4I. Only CD48^{neg} HSCs were capable of significant serial engraftment.